# Augmented Shortcuts for Vision Transformers

**Yehui Tang**[1,2], **Kai Han**[2], **Chang Xu**[3], **An Xiao**[2],
**Yiping Deng**[4], **Chao Xu**[1], **Yunhe Wang**[2]
[1]Key Lab of Machine Perception (MOE), Dept. of Machine Intelligence, Peking University.
[2]Huawei Noah's Ark Lab. [4]Central Software Institution, Huawei Technologies.
[3]School of Computer Science, Faculty of Engineering, University of Sydney.
yhtang@pku.edu.cn, {kai.han, yunhe.wang, an.xiao, yiping.deng}@huawei.com,
c.xu@sydney.edu.au, xuchao@cis.pku.edu.cn.

## Abstract

Transformer models have achieved great progress on computer vision tasks recently. The rapid development of vision transformers is mainly contributed by their high representation ability for extracting informative features from input images. However, the mainstream transformer models are designed with deep architectures, and the feature diversity will be continuously reduced as the depth increases, *i.e.*, feature collapse. In this paper, we theoretically analyze the feature collapse phenomenon and study the relationship between shortcuts and feature diversity in these transformer models. Then, we present an augmented shortcut scheme, which inserts additional paths with learnable parameters in parallel on the original shortcuts. To save the computational costs, we further explore an efficient approach that uses the block-circulant projection to implement augmented shortcuts. Extensive experiments conducted on benchmark datasets demonstrate the effectiveness of the proposed method, which brings about 1% accuracy increase of the state-of-the-art visual transformers without obviously increasing their parameters and FLOPs.

## 1 Introduction

Originating from the natural language processing field, transformer models [35, 9] have recently made great progress in various computer vision tasks such as image classification [9, 34, 23], object detection [3] and image processing [4]. Wherein, the ViT model [9] divides the input images as visual sequences and obtains an 88.36% top-1 accuracy which is competitive to the SOTA convolutional neural network (CNN) models (*e.g.*, EfficientNet [31]). Compared to CNNs which are usually customized for vision tasks with prior knowledge (*e.g.*, translation equivalence and locality), vision transformers introduce less inductive bias and have a larger potential to achieve better performance on different visual tasks. Besides, considering the high performance of transformers in various fields (*e.g.*, natural language processing and computer vision), we may only need to support transformer for processing different tasks which can significantly simplify the software/hardware design.

Besides the self-attention layers in vision transformers, a shortcut is often included to directly connect multiple layers with identity projection [15, 35, 9]. The introduce of shortcut is motivated by the architecture designs in CNNs, and has been demonstrated to be beneficial for a stable convergence and better performance [16, 33, 22, 39, 38]. There is a series of works on interpreting and understanding the role of shortcut. For example, Balduzzi *et al.* [1] analyze the gradients of deep networks and show that shortcut connection can effectively alleviate the problem of gradient vanishing and exploding. Veit *et al.* [36] reckon ResNet [15] as the ensemble of a collection of paths with different lengths, and the gradient vanishing problem is addressed by short paths. From a theoretical perspective, Liu *et al.* [22] prove that shortcut connection can avoid network parameters trapped by spurious local optimum and help them converge to a global optimum. Apart from CNNs, shortcut connection is

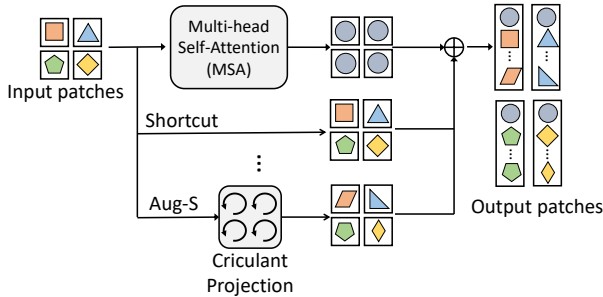
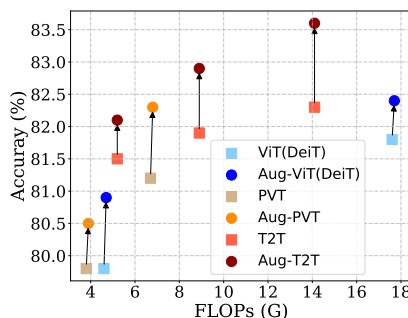

Figure 1: The diagram of MSA module equipped with augmented shortcuts, where different patterns (rectangle, triangle, etc.) denote different features from various patches. The original identity shortcut copies input feature while the augmented shortcuts (Aug-S) project features of each input patch to diverse representations.

Figure 2: Performance improvement for using the proposed augmented shortcuts on the state-of-the-art vision transformer models. The performances of different models on ImageNet are provided.

consequently widely-used in other deep neural networks such as transformer [35, 9], LSTM [11] and AlphaGo Zero [29].

In vision transformers, the shortcut connection bypasses the multihead self-attention (MSA) and multilayer perceptron (MLP) modules, which plays a critical role towards the success of vision transformers. A transformer without shortcut suffer extremely low performance (Table 1). Empirically, removing the shortcut results in features from different patches becoming indistinguishable as the network going deeper (shown in Figure 3(a)), and such features have limited representation capacity for the downstream prediction. We name this phenomenon as *feature collapse*. Fortunately, adding shortcut in transformers can alleviate the phenomenon (Figure 3(b)) and make the generated features diverse. However, the conventional shortcut simply copies the input feature to the output, limiting its ability for enhancing the feature's diversity.

In this paper, we introduce a novel augmented shortcut scheme for improving feature diversity in vision transformers (Figure 1). Besides the conventional identity shortcut, we propose to parallel the MSA module with multiple parameterized shortcuts, which provide more alternative paths to bypass the attention mechanism. In particular, an augmented shortcut connection is constructed as the sequence of a linear projection with learnable parameters and a nonlinear activation function. We theoretically demonstrate that the transformer model equipped with augmented shortcuts can avoid feature collapse and produce more diverse features. For the efficiency reason, we further replace the original dense matrices with block-circulant matrices, which have lower computational complexity in the Fourier frequency domain while high representation ability in the spatial domain. Owing to the compactness of circulant projection, the parameters and computational costs introduced by the augmented shortcuts are negligible compared to those of the MSA and MLP modules. We empirically evaluate the effectiveness of augmented shortcut with the ViT model [9, 34] and its SOTA variants (PVT [37], T2T [40]) on ImageNet dataset. Equipped with our augmented shortcut, the performances (top-1 accuracies) of these models can be enhanced by about 1% with comparable computational costs (*e.g.*, Figure 2).

## 2 Preliminaries and Motivation

The vision transformer (ViT) [9] adopts the typical architecture proposed in [35], with MSA and MLP modules alternatively stacked. For an image, ViT splits it into $N$ patches and projects each patch into a $d$-dimension vector. Thus the input of a vision transformer can be seen as a matrix $Z_0 \in \mathbb{R}^{N \times d}$. An MSA module with $H$ heads is defined as

$$\text{MSA}(Z_l) = \text{Concat}([A_{lh} Z_l W^v_{lh}]^H_{h=1}) W^o_l, \qquad (1)$$

where $Z_l \in \mathbb{R}^{N \times d}$ is the features of the $l$-th layer, $A_{lh} \in \mathbb{R}^{N \times N}$ and $W^v_{lh} \in \mathbb{R}^{d \times (d/H)}$ are the corresponding attention map and value projection matrix in the $h$-th head, respectively. $\text{Concat}(\cdot)$

denotes the concatenating for features of the $H$ heads and $W_l^o \in \mathbb{R}^{d \times d}$ is the output projection matrix. The attention matrix $A_{lh}$ is calculated by the self-attention mechanism, *i.e.*,

$$A_{lh} = \mathrm{softmax}\left(\frac{(Z_l W_{lh}^q)(Z_l W_{lh}^k)^\top}{\sqrt{d}}\right), \tag{2}$$

$$A_{lh} = \mathrm{softmax}\left(\frac{(Z_l W_{lh}^q)(Z_l W_{lh}^k)^\top}{\sqrt{d}}\right), \tag{3}$$

where $W_{lh}^q \in \mathbb{R}^{d \times (d/H)}$ and $W_{lh}^k \in \mathbb{R}^{d \times (d/H)}$ are the query and value projection matrices, respectively. Attention $A_{lh}$ reflects the relation between different patches, and a larger value $A_{lh}^{ij}$ indicate that patch $i$ and patch $j$ have a stronger relationship.

The MLP module extracts features from each patch independently, which is usually constructed by stacking two linear projection layers with weights $W^a \in \mathbb{R}^{d \times d_{hidden}}$, $W^b \in \mathbb{R}^{d_{hidden} \times d}$ and non-linear activation function $\sigma$, *i.e.*, $\mathrm{MLP}(Z_l) = \sigma(Z_l W^a)W^b$. A vision transformer model is constructed by stacking the MLP and MSA modules alternatively.

| Depth | Top1 accuracy (%) |
|-------|-------------------|
| 1 | 18.3 |
| 2 | 43.4 |
| 4 | 0.13 |
| 6 | 0.12 |
| 8 | 0.15 |
| 10 | 0.14 |
| 12 | 0.15 |

Table 1: Performance of the ViT model without shortcut on ImageNet.

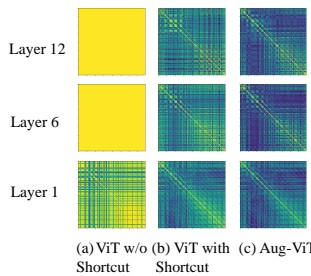

(a) ViT w/o Shortcut (b) ViT with Shortcut (c) Aug-ViT

Figure 3: The similarity matrices over patches in the ViT-Base model.

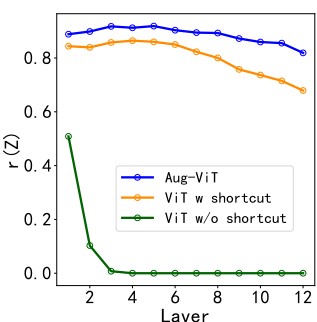

Figure 4: Diversity metric $r(Z_l)$ varies *w.r.t.* layers.

Besides MSA and MLP, the shortcut connection also plays a vital role to achieve high performance. Empirically, removing shortcut severely harms the accuracies of ViT models (as shown in Table 1). To explore the reason behind, we propose to analyze the intermediate features in the ViT-like models. Specially, we calculate the cosine similarity between different patch features and show the similarity matrices of low (Layer 1), middle (Layer 6) and deep layers (Layer 12) in Figure 3. Features from different patches in a layer quickly become indistinguishable as the network depth increasing. We call this phenomenon *feature collapse*, which greatly restrict the representation capacity and then prevent high performance.

The feature collapse mainly results from the attention mechanism, which aggregates features from different patches layer-by-layer. Denoting $\tilde{Z}_{lh}$ as the output feature after the attention map, the attention mechanism can be formulated as:

$$\tilde{Z}_{lh}^i = \sum_{j=1}^{N} A_{lh}^{ij} Z_l^j, \;\; \mathrm{s.t.} \; \sum_{j=1}^{N} A_{lh}^{ij} = 1, i = [1, 2, \cdots, N], \tag{4}$$

where the feature $\tilde{Z}_{lh}^i$ from the $i$-th patch is the weighted average of features from all other patches, and the weights are values in attention map $A_{lh}$. Though attention mechanisms capture global relationship, the diversity of patch features are reduced as well. Theoretically, the diversity of a feature $Z_l$ can be measured by the difference between the feature and a rank-1 matrix, *i.e.*,

$$r(Z_l) = \|Z_l - \mathbf{1}z_l^\top\|, \;\; \text{where } z_l = \mathrm{argmin}_{z_l'}\|Z_l - \mathbf{1}{z_l'}^\top\|. \tag{5}$$

where $\|\cdot\|$ denotes the matrix norm[1]. $z_l, z_l' \in \mathbb{R}^d$ are vectors and $\mathbf{1}$ is an all-ones vector. The rank of matrix $\mathbf{1}z_l^\top$ is 1. Intuitively, if $z_l \in \mathbb{R}^d$ can represent $Z_l \in \mathbb{R}^{N \times d}$ of $N$ patches, the generated

---

[1] Here $\|\cdot\|$ is defined as the $\ell_1, \ell_\infty$-composite norm for the convenience of theoretical derivation.

feature will be redundant. A larger $r(Z_l)$ implies stronger diversity of the given feature $Z_l$. We use $r(Z_l)$ as the diversity metric in the following. Because MSA module incurs the feature collapse, we focus on the a model stacked by attention modules and have the following theorem [8], *i.e.*,

**Theorem 1** *Given a model stacked by the MSA modules, the diversity $r(X_l)$ of feature in the l-th layer can be bounded by that of input data $Z_0$, i.e.,*

$$r(Z_l) \leq \left( \frac{H\gamma}{\sqrt{d}} \right)^{\frac{3^l-1}{2}} r(Z_0)^{3^l}, \tag{6}$$

*where $H$ is number of heads, $d$ is feature dimension and $\gamma$ is a constant related to the norms of weight matrices in the MSA module.*

$H\gamma/\sqrt{d}$ and $r(Z_0)$ are usually smaller than 1, so the feature diversity $r(Z_l)$ will decrease rapidly as the network depth increases [8]. We also empirically show how $r(Z_l)$ varies *w.r.t.* the network depth in the ViT models (Figure 4), and the empirical results are accordant to the Theorem 1.

Fortunately, adding a shortcut connection parallel to the MSA module can empirically preserve the feature diversity especially in deep layers (see Figure 3(b) and Figure 4). The MSA module with shortcut can be formulated as:

$$\text{ShortcutMSA}(Z_l) = \text{MSA}(Z_l) + Z_l, \tag{7}$$

where the identity projection (*i.e.*, $Z_l$) is parallel to the MSA module. Intuitively, the shortcut connection bypasses the MSA module and provides another alternative path, where features can be directly delivered to the next layer without interference of other patches. Adding the shortcut connections can also theoretically improve the bound of feature diversity $r(Z_l)$ (as discussed in Section 3.1). The success of shortcut shows that bypassing the attention layers with extra paths is an effective way to enhance feature diversity and improve the performance of transformer-like models.

However, in general vision transformer models [9, 13, 37, 14, 12, 40, 23], there is only a single shortcut connection with identity projection for each MSA module, which only copies the input features to the outputs. This simple formulation may not have enough representation capacity to improve the feature diversity maximally. In the following chapters, we aim to refine the existing shortcut connections in vision transformers and explore efficient but powerful augmented shortcuts to produce visual features with higher diversity.

## 3 Approach

### 3.1 Augmented Shortcuts

We propose augmented shortcuts to alleviate the feature collapse by paralleling the original identity shortcut with more parameterized projections. The MSA module equipped with $T$ augmented shortcuts can be formulated as:

$$\text{AugMSA}(Z_l) = \text{MSA}(Z_l) + Z_l + \sum_{i=1}^{T} \mathcal{T}_{li}(Z_l; \Theta_{li}), \ l \in [1, 2, \cdots, L], \tag{8}$$

where $\mathcal{T}_{li}(\cdot)$ is the $i$-th augmented shortcut connection of the $l$-th layer and $\Theta_{li}$ denotes its parameters. Besides the original shortcut, the augmented shortcuts provide more alternative paths to bypass the attention mechanism. Different from the identity projection directly copying the input patches to the corresponding outputs, the parameterized projection $\mathcal{T}_{li}(\cdot)$ can transform input features into another feature space. Actually, projections $\mathcal{T}_{li}(\cdot)$ will make different transformations on the input feature as long as their weight matrices $\Theta_{li}$ are different, and thus paralleling more augmented shortcuts has potential to enrich the feature space.

A simple formulation for $\mathcal{T}_{li}(\cdot)$ is the sequence of a linear projection and an activation function *i.e.*,

$$\mathcal{T}_{li}(Z_l; \Theta_{li}) = \sigma(Z_l \Theta_{li}), \ l \in [1, \cdots, L], \ i \in [1, 2, \cdots, T], \tag{9}$$

where $\Theta_{li} \in \mathbb{R}^{d \times d}$ is the weight matrix and $\sigma$ is the non-linear activation function (*e.g.*, GELU). In Eq. 9, $\mathcal{T}_{li}(\cdot)$ tackles each patch independently and preserves their specificity, which is complement to

the MSA modules aggregating different patches. Note that the identity mapping is a special case of Eq. 9, *i.e.*, $\sigma(x) = x$ and $\Theta_{li}$ is the identity matrix.

Recall that in a transformer-like model without shortcut, the upper bound of feature diversity $r(Z_l)$ decreases dramatically as the increase of network depth(Theorem 1). In the following, we analyze how the diversity $r(Z_l)$ changes *w.r.t.* the layer $l$ in the model stacked by the AugMSA modules, and we has the follow theorem.

**Theorem 2** *Given a model stacked by the AugMSA modules, the diversity $r(X_l)$ of feature in the l-th layer can be bounded by that of input data $Z_0$, i.e.,*

$$r(Z_l) \leq \max_{0 \leq m \leq l} \left( \frac{H\gamma}{\sqrt{d}} \right)^{\frac{3^m - 1}{2}} (2H\alpha_m)^{3^m(l-m)} r(Z_0)^{3^m}, \tag{10}$$

*where $\alpha_m = 1 + \sum_{i=1}^{T} \lambda \|\Theta_{mi}\|$. $\Theta_{mi}$ is the weight matrix in the i-th augmented shortcut of the m-th layer, and $\lambda$ is the Lipschitz constant of activation function $\sigma(\cdot)$.*

Compared with Theorem 1, the augmented shortcuts introduce an extra term $(2H\alpha_m)^{3^m(l-m)}$, which will increase doubly exponentially as $2H\alpha_m$ is usually larger than 1. This tends to suppress the diversity decay incurred by attention mechanism. The term $\alpha_m$ ($0 \leq m \leq l$) is determined by the norms of weight matrices $\Theta_{mi}$ of the augmented shortcuts in the $m$-th layer, and then bound of diversity $r(Z_l)$ in the $l$-th layer can be affected by all the augmented shortcuts in the previous layers. For the ShortcutMSA module (Eq. 7) with only a identity shortcut, we have $\alpha_m = 1$. Adding more augmented shortcuts can increase the magnitude of $\alpha_m$, which further improves the bound. Detailed proof for Theorem 2 is represented in the supplemental material.

Considering that shortcut connections exist in both MSA and MLP modules, the proposed augmented shortcuts can also be embedded into MLP similarly, *i.e.*,

$$\text{AugMLP}(Z'_l) = \text{MLP}(Z'_l) + Z'_l + \sum_{i=1}^{T} \mathcal{T}_{li}(Z'_l; \Theta'_{li}), \ l \in [1, 2, \cdots, L], \tag{11}$$

where $Z'_l$ is the input feature of the MLP module in the $l$-th layer and $\Theta'_{li}$ denotes the parameters in augmented shortcuts. Paralleling the MLP module with the augmented shortcuts can further improve the diversity, which is analyzed detailedly in the supplemental material. Stacking the AugMSA and AugMLP modules constructs the *Aug-ViT* model, whose feature has stronger diversity as shown in Figure 3(c) and Figure 4.

## 3.2   Efficient Implementation via Circulant Projection

As discussed above, Paralleling multiple augmented shortcuts with the MSA and MLP modules in a vision transformer can improve the feature diversity for higher performance. However, directly implementing $\mathcal{T}_{li}(\cdot)$ (Eq. 9) involves a lot of matrix multiplications, which are computationally expensive [19, 43, 32]. For example, given feature $Z_l \in \mathbb{R}^{n \times d}$ and weight matrix $\Theta_{li} \in \mathbb{R}^{d \times d}$, the matrix multiplication $Z_l \Theta_{li}$ consumes $nd^2$ FLOPs, where $d$ is usually large in vision transformers (*e.g.*, 798 in ViT-B). Therefore, we propose to implement the augmented shortcuts with block-circulant matrices, whose computational costs are negligible compared to other modules in a vision transformer. In the following, we omit the subscripts $l$ and $i$ indicating layers and paths for brevity.

Circulant matrix [19, 7] is a typical structured matrix with extremely few parameters and low computational complexity on the Fourier domain. A circulant matrix $C \in \mathbb{R}^{d' \times d'}$ only has $d'$ parameters and the product between $C$ and a vector only has $\mathcal{O}(d' \log d')$ computational complexity via the fast Fourier transformation (FFT). The projection with circulant matrix also has theoretical guarantee to be a good approximation of unstructured projection and preserve the critical projection properties such as $\ell_2$ distance and angles [17, 10]. Consequently, we take advantage of circulant matrices to implement augmented shortcuts. Specially, the original matrix $\Theta$ is split into $b^2$ sub-matrices $C^{ij} \in \mathbb{R}^{d' \times d'}$, *i.e.*,

$$\Theta = \begin{bmatrix} C^{11} & C^{12} & \cdots & C^{1b} \\ C^{21} & C^{22} & \cdots & C^{2b} \\ \vdots & \vdots & \vdots & \vdots \\ C^{b1} & C^{b2} & \cdots & C^{bb} \end{bmatrix}, \tag{12}$$

where $d'$ is the size of sub-matrices and $bd' = d$. Each sub-matrix $C^{ij}$ is a circulant matrix generated by circulating the elements in a $d'$-dimension vector $\boldsymbol{c}^{ij} = [c_1^{ij}, c_2^{ij}, \cdots, c_{d'}^{ij}]$, i.e.,

$$C^{ij} = circ(\boldsymbol{c}^{ij}) = \begin{bmatrix} c_1^{ij} & c_{d'}^{ij} & \cdots & c_3^{ij} & c_2^{ij} \\ c_2^{ij} & c_1^{ij} & c_{d'}^{ij} & & c_3^{ij} \\ \vdots & c_2^{ij} & c_1^{ij} & \ddots & \vdots \\ c_{d'-1}^{ij} & & \ddots & \ddots & c_d^{ij} \\ c_{d'}^{ij} & c_{d'-1}^{ij} & \cdots & c_2^{ij} & c_1^{ij} \end{bmatrix}. \tag{13}$$

For efficient implementation of the projection $\mathcal{T}(Z) = \sigma(Z\Theta)$, we first split the input $Z$ into $b$ slices $Z^j \in \mathbb{R}^{N \times d'}$, i.e., $Z = [Z^1; Z^2; \cdots; Z^b]$, and then multiply each slice $Z^j$ by a circulant matrix $C^{ij}$. The product between circulant matrix and vector in the original domain is equivalent to element-wise multiplication in the Fourier domain, i.e., $\mathcal{F}(Z^j C^{ij}) = \mathcal{F}(Z^j) \odot \mathcal{F}(\boldsymbol{c}^{ij})$, where $\mathcal{F}(\cdot)$ is the discrete Fourier transformation and $\odot$ denotes that each row in $Z^j$ is multiplied by the according element in $\boldsymbol{c}^{ij}$. The discrete Fourier transform and its inverse transformation both can be efficiently calculated with fast Fourier transform FFT and its inverse IFFT with only $\mathcal{O}(d' \log d')$ computational complexity. The output is calculated as:

$$\mathcal{T}(Z)^i = \sigma\left(\sum_{j=1}^{b} Z^j C^{ij}\right) = \sigma\left(\sum_{j=1}^{b} \text{IFFT}(\text{FFT}(Z^j) \circ \text{FFT}(\boldsymbol{c}^{ij}))\right), \tag{14}$$

where $\mathcal{T}(Z)^i \in \mathbb{R}^{N \times d'}$ is a slice of $\mathcal{T}(Z)$. Finally, $\mathcal{T}(Z)$ is obtained by concatenating slices $\mathcal{T}(Z)^i$, i.e., $\mathcal{T}(Z) = [\mathcal{T}(Z)^1; \mathcal{T}(Z)^2; \cdots; \mathcal{T}(Z)^b]$.

**Complexity Analysis.** For a $d \times d$ matrix $\Theta$ split into multiple $b^2$ sub-matrices, the numbers of parameters is only $bd$, which has linear complexity with matrix size $d$. Note that the learnable parameters $\boldsymbol{c}^{ij}$ can be stored as its Fourier form, and both the FFT and IFFT operations in Eq. 14 are conducted one time. Thus the computational cost (i.e., FLOPs) of Eq. 14 is about $(d \log(d/b) + 2bd)$, where $d \log(d/b)$ comes from the FFT and IFFT transformation and $2bd$ is consumed by the element-wisely operation in the Fourier domain [26, 2][2]. The size of matrix $d$ in a transformer is usually large while the number of sub-matrices $b$ is small, and thus the parameters and computation costs incurred by the augmented shortcuts can be negligible. For example, in the ViT-S model with $d = 384$, $b$ is set to 4 in our implementation. The augmented shortcuts only adds 0.07 M parameters and 0.08 G FLOPs, which is negligible considering the ViT-S model has 22.1 M parameters and 4.6 G FLOPs.

## 4 Experiments

In this section, we conduct extensive experiments to demonstrate the effectiveness of the proposed augmented shortcuts. The vision transformer [9] and its SOTA variants are equipped with the augmented shortcuts for performance improving. We first compare the performances of different models on the ImageNet dataset for the image classification task. Then ablation studies are conducted to analyze the algorithm. To validate its generalization ability, we further test the proposed Aug-ViT model on the object detection and transfer learning tasks.

### 4.1 Experiments on ImageNet

**Dataset.** ImageNet (ILSVRC-2012) dataset [6] contains 1.3 M training images and 50k validation images from 1000 classes, which is a widely used image classification benchmark.

**Implementation details.** We use the same training strategy with DeiT [34] for a fair comparison. Specifically, the model is trained with AdamW [24] optimizer for 300 epochs with batchsize 1024. The learning rate is initialized to $10^{-3}$ and then decayed with the cosine schedule. Label smoothing [30], DropPath [21] and repeated augmentation [18] are also implemented following DeiT [34]. The data augmentation strategy contains Rand-Augment [5], Mixup [42] and CutMix [41]. Besides the original

---

[2]The constants are approximated by considering the relation between complex operation and real operation, as well as the symmetry property [2]

shortcuts, one or two augmented shortcuts are added, where the hyper-parameter $b$ for partitioning matrices is set to 4 empirically. The models are trained from scratch on ImageNet and no extra data are used. All experiments are conducted with PyTorch [28] and MindSpore [3] on NVIDIA V100 GPUs. The model architecture can be found in the MindSpore model zoo [4].

**Backbones.** We apply the augmented shortcuts on multiple vision transformer models. ViT [9] is the typical transformer model for the vision tasks, which splits an image to multiple $16 \times 16$ patches. Deit [34] adopts the same architecture with ViT but improves the training strategy for better performance. T2T [40] and PVT [37] are two recently proposed SOTA variants of ViT. T2T [40] improves the process of producing patches by considering the structured information in images. PVT [37] designs a pyramid-like structure by partitioning the model into multiple stages. In Table 2, 'ViT(DeiT)-S' ,'ViT(DeiT)-B' and 'PVT-S', 'PVT-B' denote the DeiT and PVT models with different size. 'T2T-14', 'T2T-19' and 'T2T-24' are the T2T models with different depths.

**Experimental Results.** The validation accuracies of different models on ImageNet are shown in Table 2. [5] We firstly equip the ViT models with the augmented shortcuts and get Aug-ViT, which show large superiority to the plain counterparts, *i.e.*, more than 1% accuracy improvement without noticeable parameter and computational complexity increasing. For example, the proposed 'Aug-ViT-S' achieves 80.9% top-1 accuracy with 4.6G FLOPs, which suppresses the baseline ('ViT(DeiT)-S' with 4.6G FLOPs) by 1.2% top-1 accuracy while the computational cost is barely changed. For a large model with higher input image resolution (*e.g.*, ViT-B with input resolution $384 \times 384$) and high performance, equipping it with the augmented shortcuts can still further improve the performance (*e.g.*, 83.1% $\rightarrow$ 84.2%).

Besides the typical ViT model, the augmented shortcuts can be embedded into multiple vision transformers flexibly and improve their performance as well. For example, equipping the PVT-M with augmented shortcut can improve its accuracy form 81.2% to 82.3%. For the T2T-14 model, the performance improvement is even more obvious, *i.e.*, 1.3% accuracy improvement from 82.3% to 83.6%.

**Performance Improvement *w.r.t.* Network Depth.** It is interesting to see that the augmented shortcuts improve the performance of deeper models more obviously. 'T2T-14', 'T2T-19' and 'T2T-24' compose of the same blocks but have different depths. For the baseline, increasing the depth of T2T model from 14 to 24, the accuracy is only improved by 0.7% (from 81.5% to 82.3%). While with the augmented shortcuts, the Aug-T2T model with 24 layers can achieve 83.6%, which achieves more obvious performance improvement than models with 14 layers. Deeper models tend to suffer more serve feature collapse suffer more serve feature collapse as features from different patches are aggregated with more attention layers. The proposed augmented shortcuts alleviate the feature collapse and release the potential of deep models, which results in higher performance.

## 4.2  Ablation Studies

To better understand the proposed augmented shortcuts for vision transformers, we conduct extensive experiments to investigate the impact of each component. All the ablation experiments are conducted based on ViT(DeiT)-S model on the ImageNet dataset.

**The number of augmented shortcuts.** The performance varies *w.r.t.* the number of augmented shortcuts as shown in Table 3. Besides the original identity shortcut, adding only one augmented shortcut can significantly improves the performance of the ViT model (*e.g.*, 0.8% top-1 accuracy improvement compared to the baseline). Further increasing the number of augmented shortcuts will further improve the performance, and the improvement margin will be saturated gradually. We empirically find that two augmented shortcuts are enough to achieve obvious performance improvement.

**Location for implementing the augmented shortcuts.** As discussed before, the augmented shortcuts can be paralleled with both MSA and MLP modules to increase the feature diversity. Table 4 shows how the implementation location affects the final performance. Paralleling MSA with the

---

[3]`https://www.mindspore.cn`

[4]`https://gitee.com/mindspore/models/tree/master/research/cv/augvit`

[5]Note that DeiT [34] adopts the same model architecture with ViT [9] but achieves higher performance by adjusting the training strategy, which is used as the baseline model.

Table 2: Performance of different models on ImageNet.

| Model | Resolution | Top1-Accuracy (%) | Params (M) | FLOPs (G) |
|---|---|---|---|---|
| ViT(DeiT)-S [9] | 224×224 | 79.8 | 22.1 | 4.6 |
| Aug-ViT-S | 224×224 | **80.9** (+1.1) | 22.2 | 4.7 |
| ViT(DeiT)-B [9] | 224×224 | 81.8 | 86.4 | 17.6 |
| Aug-ViT-B | 224×224 | **82.4** (+0.6) | 86.5 | 17.7 |
| ViT(DeiT)-B↑ [9] | 384×384 | 83.1 | 86.4 | 55.6 |
| Aug-ViT-B↑ | 384×384 | **84.2** (+1.1) | 86.5 | 55.8 |
| PVT-S [37] | 224×224 | 79.8 | 24.5 | 3.8 |
| Aug-PVT-S [37] | 224×224 | **80.5** (+0.7) | 24.6 | 3.9 |
| PVT-M [37] | 224×224 | 81.2 | 44.2 | 6.7 |
| Aug-PVT-M [37] | 224×224 | **82.3** (+1.1) | 44.3 | 6.8 |
| T2T-14 [40] | 224×224 | 81.5 | 21.5 | 5.2 |
| Aug-T2T-14 | 224×224 | **82.1** (+0.6) | 21.6 | 5.3 |
| T2T-19 [40] | 224×224 | 81.9 | 39.2 | 8.9 |
| Aug-T2T-19 | 224×224 | **82.9** (+1.0) | 39.3 | 9.0 |
| T2T-24 [40] | 224×224 | 82.3 | 64.1 | 14.1 |
| Aug-T2T-24 | 224×224 | **83.6** (+1.3) | 64.2 | 14.3 |

Table 3: The number of augmented shortcuts.

| # Path | Top1-Acc. (%) | Params (M) | FLOPs (G) |
|---|---|---|---|
| 0 | 79.8 (+0.0) | 22.1 | 4.6 |
| 1 | 80.6 (+0.8) | +0.04 | +0.04 |
| 2 | 80.9 (+1.1) | +0.07 | +0.08 |
| 3 | 80.9 (+1.1) | +0.12 | +0.12 |

Table 4: Location for implementing the augmented shortcuts.

| Location | Top1-Acc. (%) | Params (M) | FLOPs (G) |
|---|---|---|---|
| Baseline | 79.8 (+0.0) | 22.1 | 4.6 |
| MSA | 80.6 (+0.8) | +0.035 | +0.04 |
| MLP | 80.4 (+0.6) | +0.035 | +0.04 |
| MSA&MLP | 80.9 (+1.1) | +0.07 | +0.08 |

augmented shortcuts significantly improves the performance (*e.g.*, 0.8% top-1 accuracy), which we attribute it to the increasing of feature diversity. Enhancing MLP module also bring the performance improvement, which is accordant to our analysis in supplementary materials. Combining them together can achieve the highest performance (1.1% accuracy improvement), which we adopt in our implementation.

**Efficiency of the block-circulant projection.** In the block-wisely circulant projection, the hyper-parameter $b$ controls the number of sub-matrices $C^{ij}$ partitioned by original matrix $\Theta$. Table 5 shows how the performance varies *w.r.t.* the parameter $b$. A larger $b$ implies the matrix $\Theta$ will be partitioned into more circulant matrix with smaller sizes, which brings more parameters and higher computational cost, as well as the performance improvement. The unstructured projection can also be used as augmented shortcuts and brings performance improvement, but it incurs obvious increasing of parameters and computational cost. Using the block-circulant projection with $b = 4$ can achieve very similar performance with the unstructured projection but has much few parameters (*e.g.*, 0.07M *vs.* 7.1M), implying that the block-circulant projection is an efficient and effective formulation to transform features in the augmented shortcuts.

**Formulation of the augmented shortcuts.** The augmented shortcuts are implemented sequentially with the block-circulant projection and activation function (*e.g.*, GeLU). Table 6 shows the impact of each component. Only using the activation function introduces no learnable parameters, which only achieves similar accuracy with the baseline model, implying that the learnable parameters are vital to produce diverse features. If only the linear circulant projection is kept, multiple augmented shortcut can be merged to a single one. Swapping the sequential order of the circulant projection and activation function has negligible influence on the final performance.

**Drop rates of stochastic depth and dropout.** We change the parameters of stochastic depth/dropout for both ViT(DeiT)-S model and Aug-ViT-S, and the results are shown in Table 7 and Table 8. We empirically find the default parameters for DeiT (drop rates of stochastic depth/Dropout are set as 0.1/0.0) also work well for the Aug-VIT models. In addition, the models equipped with augmented shortcuts tend to have higher tolerance to over-large drop rates. For example, with drop rate=0.3 for

Table 5: Efficiency of the block cifculant projection.

| Type | $b$ | Top1-Acc. (%) | Params (M) | FLOPs (G) |
|---|---|---|---|---|
| Baseline | - | 79.8 | 22.1 | 4.6 |
| Block-Circulant | 1 | 80.5 (+0.7) | +0.02 | +0.07 |
| Block-Circulant | 2 | 80.7 (+0.9) | +0.03 | +0.07 |
| Block-Circulant | 4 | 80.9 (+1.1) | +0.07 | +0.08 |
| Block-Circulant | 8 | 80.9 (+1.1) | +0.15 | +0.10 |
| Unstructured | - | 81.0 (+1.2) | +7.1 | +1.4 |

Table 6: Formulation of the augmented shortcuts. 'Act' denotes the activation function.

| Form | Top1-Acc. (%) |
|---|---|
| Baseline | 79.8 |
| Act. | 79.9 |
| Circulant | 80.6 |
| Act. + Circulant | 80.8 |
| Circulant + Act. | 80.9 |

Table 7: Top-1 accuracies (%) on ImageNet *w.r.t.* drop rates of stochastic depth.

| Drop rate | DeiT-S | Aug-ViT-S |
|---|---|---|
| 0 | 79.7 | 80.9 |
| 0.05 | 79.8 | 80.9 |
| 0.1 | 79.8 | 80.9 |
| 0.2 | 79.3 | 80.4 |
| 0.3 | 78.6 | 80.0 |

Table 8: Top-1 accuracies (%) on ImageNet *w.r.t.* drop rates of dropout.

| Drop rate | DeiT-S | Aug-ViT-S |
|---|---|---|
| 0 | 79.8 | 80.9 |
| 0.05 | 78.9 | 80.3 |
| 0.1 | 77.3 | 79.1 |
| 0.2 | 74.0 | 76.5 |
| 0.3 | 69.8 | 75.1 |

dropout, the accuracy of Aug-ViT-S can still achieve 75.1% accuracy, while the accuracy of DeiT-S decreases to 69.8%.

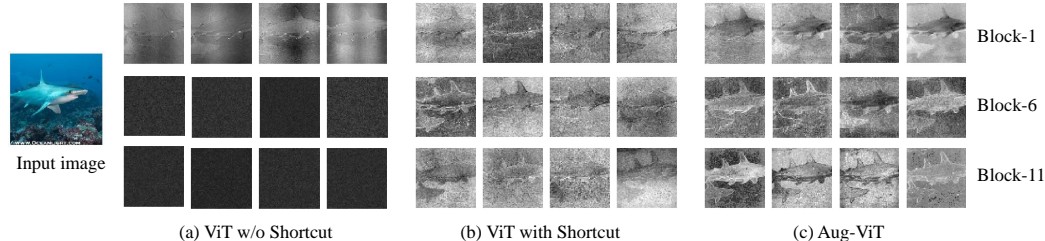

Figure 5: Feature Visualization of different models.

**Feature Visualization.** We intuitively show the features of different models in Figure 5, From top to bottom are features in low, middle and deep layers of the ViT-Small model. The input image is scaled to $1024 \times 1024$ for better visualization and the patch embeddings are reshaped to their spatial positions to construct the features maps. Without shortcut, the feature maps in deep layers conveys no effective information (Figure 5 (a)), and adding a shortcut connection make the feature maps informative ((b)). Compared with them, the features in the Aug-ViT model are further enriched, especially for the deep layers.

### 4.3   Object Detection with Pure Transformer

We also validate the effectiveness of the augmented shortcuts on the object detection task. A pure transformer detector can be constructed by combining the vision transformer backbone and the DETR [3] head, and we equip the backbones with the augmented shortcuts. For a fair comparison, we follow the training strategy in PVT [37] and fine-tune the models for 50 epochs on the COCO train2017 dataset. Random flip and random scale are used as the data augmentation strategy. The results on COCO val2017 are shown in Table 9. The detector equipped with augmented shortcut achieve better performance than the base model. For the DeiT-S model with 33.9 AP, the augmented shortcut can improve 1.8% AP and achieve 35.7%.

### 4.4   Transfer Learning

To validate the generalization ability of vision transformers equipped with the projection, we conduct experiments on the transfer learning tasks. Specifically, the models trained on ImageNet is further

Table 9: Results of pure transformer object detection on COCO val2017 dataset.

| Backbone | Parmas(M) | Epochs | AP | $AP_{50}$ | $AP_{75}$ | $AP_S$ | $AP_M$ | $AP_L$ |
|---|---|---|---|---|---|---|---|---|
| Backbone | 41 | 50 | 32.3 | 53.9 | 32.3 | 10.7 | 33.8 | 53.0 |
| DeiT-S [34] | 38 | 50 | 33.9 | 54.7 | 34.3 | 11.0 | 35.4 | 56.6 |
| Aug-ViT-S | 38 | 50 | **35.7** | **56.7** | **36.5** | **12.2** | **38.0** | **58.6** |
| PVT-S [37] | 40 | 50 | 34.7 | 55.7 | 35.4 | 12.0 | 36.4 | 56.7 |
| Aug-PVT-S | 40 | 50 | **35.1** | **55.8** | **35.9** | **12.5** | **36.5** | **57.3** |

fine-tuned on the downstream tasks, containing superordinate-level image recognition dataset (CIFAR-10 [20], CIFAR-100 [20]) and fine-grained image recognition dataset( Oxford 102 Flowers [25] and Oxford-IIIT Pets [27]). Following [34], we fine-tune the models on images with resolution $384 \times 384$, and use the same fine-tuning strategy as [34]. Table 10 shows the performances of different models on the downstream tasks, and the model equipped with the augmented shortcut always achieves higher accuracies than the baseline on different tasks.

Table 10: Results on downstream tasks with ImageNet pre-training. All the models are fine-tuned with the image resolution $384 \times 384$.

| Model | Parmas(M) | ImageNet | CIFAR-10 | CIFAR-100 | Flowers | Pets |
|---|---|---|---|---|---|---|
| ViT-B/16$_{\uparrow 384}$ | 86.4 | 77.9 | 98.1 | 87.1 | 89.5 | 93.8 |
| DeiT-B$_{\uparrow 384}$ | 86.4 | 83.1 | 99.1 | 90.8 | 98.4 | - |
| Aug-ViT-B$_{\uparrow 384}$ | **86.4** | **84.1** | **99.2** | **91.3** | **98.8** | **95.1** |

## 5 Conclusion

We presented augmented shortcuts for resolving the feature collapse issue in vision transformers. The augmented shortcuts are parallel with the original identity shortcuts, and each connection has its own learnable parameters to make diverse transformations on the input features. Efficient circulant projections are used to implement the augmented shortcuts, whose memory and computational cost are negligible compared with other components in the vision transformer. Similar to the widely used identity shortcuts, the augmented shortcuts do not depend on the specific architecture design either, which can be flexibly embedded into various variants of vision transformers (*e.g.*, ViT [9], T2T [40], PVT [37]) for enhancing their performance on different tasks such as image classification, object detection and transfer learning. In the future, we plan to research designing deeper vision transformers with the help of augmented shortcuts.

## Acknowledgment

This work is supported by National Natural Science Foundation of China under Grant No. 61876007, and Australian Research Council under Project DE180101438 and DP210101859.

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
