# Augmented Shortcuts for Vision Transformers (Supplementary Material)

**Yehui Tang**[1,2], **Kai Han**[2], **Chang Xu**[3], **An Xiao**[2],
**Yiping Deng**[4], **Chao Xu**[1], **Yunhe Wang**[2]
[1]Key Lab of Machine Perception (MOE), Dept. of Machine Intelligence, Peking University.
[2]Noah's Ark Lab, Huawei Technologies. [4]Central Software Institution, Huawei Technologies.
[3]School of Computer Science, Faculty of Engineering, University of Sydney.
yhtang@pku.edu.cn, {kai.han, yunhe.wang, an.xiao, yiping.deng}@huawei.com,
c.xu@sydney.edu.au, xuchao@cis.pku.edu.cn.

## S1 Augmented shortcuts for MSA

In this section, we analyze the feature diversity in a model stacked by AugMSA modules and prove Theorem 2 in the main paper.

Supposing $z_l$ is the vector satisfying the definition of diversity $r(Z_l)$ (Eq (4) in the main paper) in the $l$-th layer, the diversity $r(\mathcal{T}_{li}(Z_l))$ of feature outputted by the augmented shortcut $\mathcal{T}_{li}(\cdot)$ can be bounded as:

$$r(\mathcal{T}_{li}(Z_l)) \leq \left\| \mathcal{T}_{li}(Z_l) - \mathcal{T}_{li}(\mathbf{1}z_l^\top) \right\| = \left\| \sigma(Z_l\Theta_{li}) - \sigma(\mathbf{1}z_l^\top\Theta_{li}) \right\|, \tag{S.1}$$

where the inequality comes from the definition of feature diversity (Eq (4) in the main paper). Taking advantage of Lipschitz continuity [3, 1] of the linear projection and non-linear activation function (*e.g.*, GLUE), the bound can be further written as:

$$r(\mathcal{T}_{li}(Z_l)) \leq \lambda \left\| \Theta_{li} \right\| \left\| Z_l - \mathbf{1}z_l^\top \right\| = \lambda \left\| \Theta_{li} \right\| r(Z_l), \tag{S.2}$$

where $\lambda$ is the Lipschitz constant of non-linear activation function and $\left\| \Theta_{li} \right\|$ is the norm of weight matrix $\Theta_{li}$. The above inequation shows that the diversity of output feature $\mathcal{T}_{li}(Z_l)$ can be bounded as that of input feature $Z_l$. Combining it with a single head attention module (Lemma A.1 in [2]) and the original identity shortcut, the diversity $r(Z_{l+1})$ after the AugMSA module is bounded as:

$$r(Z_{l+1}) \leq \frac{\gamma'}{\sqrt{d}}r(Z_{l-1})^3 + (1 + \sum_{i=1}^{T}\lambda\left\|\Theta_{li}\right\|)r(Z_l), \tag{S.3}$$

where $\gamma'$ is a constant related to the weights in the MSA module. Considering that H heads exist in the MSA module, the diversity $r(Z_{l+1})$ becomes:

$$r(Z_{l+1}) \leq \frac{H\gamma'}{\sqrt{d}}r(Z_{l-1})^3 + H(1 + \sum_{i=1}^{T}\lambda\left\|\Theta_{li}\right\|)r(Z_l) \leq \max(\frac{H\gamma}{\sqrt{d}}r(Z_{l-1})^3, 2H\alpha_l r(Z_{l-1})), \tag{S.4}$$

where $\alpha_l = 1 + \sum_{i=1}^{T}\lambda\|\Theta_{li}\|$ and $\gamma = 2\gamma'$. The above inequation can be unrolled to reflect how feature diversity varies in the whole model. From the input feature $Z_0$ to the current feature $Z_l$, one of the two terms (*i.e.*, $\frac{H\gamma}{\sqrt{d}}r(Z_{l-1})^3$ and $2H\alpha_l r(Z_{l-1})$) will be selected as the maximum value in each layer, and then we have

$$r(Z_l) \leq \max_{0 \leq m \leq l}\left(\frac{H\gamma}{\sqrt{d}}\right)^{\frac{3^m-1}{2}}(2H\alpha_m)^{3^m(l-m)}r(Z_0)^{3^m}. \tag{S.5}$$

The above inequation matches the statement of Theorem 2 in the main paper.

35th Conference on Neural Information Processing Systems (NeurIPS 2021).

## S2 Augmented shortcuts for MLP

Following that original shortcut connections exist in both MSA and MLP modules, the proposed augmented shortcuts are also embedded into the MLP module (Eq. 10 in the main paper). In this section, we analyze how the feature diversity varies in the Aug-ViT model stacked by AugMSA and AugMLP modules.

The intermediate features of the Aug-ViT model are denoted as $Z_l$ and $Z_l'$, which are the input feature of MSA and MLP modules, respectively. Supposing $\boldsymbol{z}_l'$ is the vector satisfying the definition of diversity $r(Z_l')$ (Eq (4) in the main paper) in the $l$-th layer, the diversity $r(Z_{l+1})$ after the AugMLP module can be bounded as:

$$r(Z_{l+1}) \leq \left\| \text{AugMLP}(Z_l') - \text{AugMLP}(\mathbf{1}\boldsymbol{z}_l'^{\top}) \right\| \leq (\delta + \beta_l) \left\| Z_l' - \mathbf{1}\boldsymbol{z}_l'^{\top} \right\| = (\delta + \beta_l)r(Z_l'), \quad \text{(S.6)}$$

where the first inequality comes from the definition of diversity $Z_{l+1}$ and the second is from Lipschitz continuity of the AugMLP module. Considering that the AugMLP module composes of linear projections and non-linear activation functions (*e.g.*, GLUE), it satisfies the Lipschitz continuity [3, 1]. The Lipschitz constant can be approximated as $(\delta + \beta_l)$, where $\delta$ is the Lipschitz constant of MLP module and $\beta_l = 1 + \sum_{i=1}^{T} \lambda \|\Theta_{li}'\|$ is related to the augmented shortcuts [3]. Combining the bound of feature diversity in the AugMSA module (Eq. S.4), the diversity in a block of Aug-ViT model is:

$$r(Z_{l+1}) \leq (\delta + \beta_l) \max \left( \frac{H\gamma}{\sqrt{d}} r(Z_{l-1})^3, 2H\alpha_l r(Z_{l-1}) \right). \quad \text{(S.7)}$$

Then for the whole Aug-ViT model, we have the following theorem:

**Theorem S.1** *Given a Aug-ViT model, the diversity $r(Z_l)$ of feature in the $l$-th layer can be bounded by that of input data $Z_0$, i.e.,*

$$r(Z_l) \leq \max_{0 \leq m \leq l} \left( \frac{H\gamma(\delta + \beta_m)}{\sqrt{d}} \right)^{\frac{3^m - 1}{2}} (2H\alpha_m(\delta + \beta_m))^{3^m(l-m)} r(Z_0)^{3^m}, \quad \text{(S.8)}$$

*where $\delta$ is the Lipschitz constant for the MLP module, and $\beta_m = 1 + \sum_{i=1}^{T} \lambda \|\Theta_{mi}'\|$ is introduced by the augmented shortcuts.*

The above equation shows that paralleling the MLP module with the augmented shortcuts tend to further improve the diversity, as $(\delta + \beta_m)$ is usually larger than 1. Intuitively, the augmented shortcuts enhance the representation ability of the MLP module and a more powerful MLP tends to produce more diverse features for each patch, as MLP processes different patch independently. However, it is indirect as the feature produced by the AugMLP module will still be aggregated by attention mechanisms. Hence we use the augmented shortcuts for MSA as the main components and those for MLP as assistants.