# OpenReview forum: "Augmented Shortcuts for Vision Transformers"
_NeurIPS.cc/2021/Conference — NeurIPS 2021 Poster_

### Official Review · Reviewer_pv6P · 2021-07-14

**Rating:** 8
**Confidence:** 5

**Summary:**

This paper introduce a universal module to improve vision transformers. They show that vision transformers suffer feature collapse, which mean that features in deep layers will become similar. To address, this paper presents an augmented shortcut module to  be flexibly embedded into various transformer models. Extensive experiments are conducted on multiple tasks (image classification, object detection, and transfer learning), which show that augmented shortcuts can bring obvious performance improvements. That is a good paper to me because of the new and effective module which is a new investigation to the field.

**Ethics Review Area:**

["I don’t know"]

**Limitations And Societal Impact:**

- Figure 1 illustrates how to equip the MSA module with the proposed augmented shortcuts. What do patterns (rectangle, triangle, etc.) denote? Do the patterns with different colors denote different features from various patches? It will be more intuitive to add more detailed captions.

- The parameter b in block-wise circulant matrix can affect its computational cost. How do you determine its value? Is the parameter b sensitive to different architectures of vision transformers?

[1] Deep Residual Learning for Image Recognition, CVPR 2016
[2] Densely Connected Convolutional Networks, CVPR 2017


**Main Review:**


+ The proposed method is novel and reasonable and extend the vision transformer from a new viewpoint and new investigation into the it. The augmented shortcuts can effectively alleviate feature collapse and enhance the representational ability, extensively validated both theoretically and empirically. The introduced block-wise circulant matrix can effectively improve computational efficiency.

+ The proposed augmented shortcut module has favorable flexibility and can be implemented on different transformer models for various tasks. Their module is generic and can be compatible with different transformer architectures on various models (e.g., ViT and SOTA variants PVT, T2T), which show obvious accuracy improvements. The commonality of the augmented shortcut make it have a lot of application scenarios.


**Time Spent Reviewing:**

11

---

> ### Author Response · Authors · 2021-08-10
> **Our response to Reviewer pv6P**
>
> Thanks for your constructive comments.
>
> **Q1**: The captions in Figure 1.
>
> **A1**: Thanks for your suggestion. Your understanding is correct, i.e., different patterns (rectangle, triangle, etc.) in Figure 1 do denote different features from various patches. We will add the detailed captions to Figure 1 in the final version.
>
> **Q2**: The parameter b in block-wise circulant matrix.
>
> **A2**: A larger b brings higher computational cost and improves performance as well. We empirically find that a small b (e.g., 4) can already bring obvious performance improvement.  The parameter b is robust for different architectures of vision transformers, and we set b=4 for all the vision transformers in our experiments.

---

### Official Review · Reviewer_UjMK · 2021-07-15

**Rating:** 7
**Confidence:** 5

**Summary:**

This paper alleviates feature collapse in vision transformers by introducing an efficient module, named augmented shortcut. As shown in the theoretical and empirical results, the feature diversity is enhanced and obvious performance improvement is achieved. The authors conduct experiments with various SOTA vision transformer models and validate the effectiveness of the proposed augmented shortcuts.

**Ethical Concerns:**

These aspects have been well discussed.

**Limitations And Societal Impact:**

These aspects have been well discussed.

**Main Review:**

Positive:

The proposed method is interesting and has a clear motivation. The authors observe that massive features in vision transformers are very similar, which may restrict their representation capacity. It is interesting to solve the issue by improving the shortcuts. Multiple paths with parameters have stronger power to preserve the feature diversity.

The block-wise circulant projection makes the augmented shortcut can be calculated very efficiently. The extra computation cost brought by the augmented shortcuts can be neglected. Thus the augmented shortcut can be flexibly implemented various models, just as the identity shortcut.

Extensive experiments are conducted with various models (including ViT, PVT, T2T-ViT) on different mainstream tasks. For the classification task on ImageNet, the augmented shortcuts improve the performance by 1% top-1 accuracy, which is significant. For the detection task, all the metrics are improved. These experimental results validate the superiority of the proposed method.

The components and hyper-parameters of the proposed method have been well analyzed with the ablation studies. It is glad to see that the proposed method is robust to these hyper-parameters.

Negative:

Making more discussions about the experimental results in Table 2 will make the paper stronger. For example, it is interesting to see that the augmented shortcut can improve the performance of the deeper model more significantly. Could the authors make more analyses about the reason behind it?

The captions in Figure 3 are too small to read. The authors are required to refine them.


**Time Spent Reviewing:**

4 hours

---

> ### Author Response · Authors · 2021-08-10
> **Our response to Reviewer UjMK**
>
> Thanks for your constructive comments.
>
> **Q1**: More discussions about the experimental results in Table 2.
>
> **A1**: As shown in Figure 4 of the main paper, the feature diversity reduces with the increase of model depth. Thus deeper vision transformers tend to suffer feature collapse more severely, which restricts their representation capacities. The proposed augmented shortcuts alleviate the feature collapse and release the potential of deep models, which results in higher performance.
>
> **Q2**: The captions in Figure 3 are too small to read.
>
> **A2**: Thanks for your suggestion. We will refine it in the final version.

---

> > ### Comment · Reviewer_UjMK · 2021-09-08
> > **Thanks**
> >
> > Thanks for addressing my concerns. I am really satisfied with the paper, so I keep my initial score of ACCEPT.  Please carefully revise the paper according to the comments.

---

### Official Review · Reviewer_E6ty · 2021-07-16

**Rating:** 7
**Confidence:** 5

**Summary:**

This paper introduces a simple, effective and efficient way to address the feature collapse issue in current Vision Transformer models, which is add multiple parametrized shortcut paths in parallel to the original identity shortcut path, with an efficient approximation implementation. Experiments show the proposed method consistently improves over the baseline and the authors show transfer to detection task as well.

**Limitations And Societal Impact:**

The authors have properly addressed the limitations.

**Main Review:**


The idea to use multiple parameterized shortcut to reduce the feature collapse is straightforward and effective, shown both theoretically and empirically. Actually, it is a bit supervising to see it works so well, e.g. 1.0% improvement on ImageNet by simply adding more shortcuts.

The paper is extremely well structured, and easy to follow. The experiments are extensive and ablation studies are comprehensive to understand the proposed method.

The paper proposes to add additional shortcut paths, I’m wondering what’s the performance like if simply replace the original identify shortcut to parameterized one (1x1 shortcut)?

There is a value projection inside the attention calculation, what if we simply use this projection output as augmented shortcut, what would the performance be like? in this way, no extra params would be introduced, and we get a parametrized shortcut as well.

---------------Post Rebuttal Feedback-------------

The authors' response make sense to me and solved my questions. I'll keep my rating.



**Time Spent Reviewing:**

3

---

> ### Author Response · Authors · 2021-08-10
> **Our response to Reviewer E6ty**
>
> Thanks for your constructive comments and suggestions.
>
> **Q1**: What’s the performance like if simply replace the original identify shortcut to parameterized one (1x1 shortcut)?
>
> **A1**: We empirically find that simply replacing the identify shortcut to 1x1 shortcut will harm the performance. For example, the top-1 accuracy of DeiT-S model will decrease from 79.8% to 68.4% on ImageNet, which indicates that the identity shortcut cannot be simply removed. We conjecture that the identity shortcut is important to optimize deep models as it can provide an extremely short path to propagate gradients [r1]. Thus we keep the identity shortcuts and add augmented shortcuts to enhance feature diversity.
>
> **Q2**: Use the value projection output as augmented shortcut.
>
> **A2**: Thanks for your suggestion. This is a very interesting proposal. The results of using the value projection (Aug-ViT-S (value)) as the augmented shortcut are shown in Table R1. Aug-ViT-S (value) achieves top-1 accuracy of 80.0%, which is higher than that of ViT(DeiT)-S (79.8%). However, it is still inferior to the performance of Aug-ViT-S, and we conclude that the parameters sharing between value projection and augmented shortcut may impede the improvement of feature diversity.
>
> Table R1: Performance of ViT(DeiT)-S, Aug-ViT-S (value) and Aug-ViT-S on ImageNet.
>
> | Model             | Top-1 Accuracy (%) |
> | ----------------- | ------------------ |
> | ViT(DeiT)-S       | 79.8               |
> | Aug-ViT-S (value) | 80.0 (+0.2)        |
> | Aug-ViT-S         | 80.9 (+1.1)        |
>
>  [r1] Identity Mappings in Deep Residual Networks, ECCV 2016.

---

### Official Review · Reviewer_1wm5 · 2021-07-17

**Rating:** 6
**Confidence:** 4

**Summary:**

This work proposes to boost the performance of vision transformers from shortcuts. As analyzed, the feature diversity of tokens declines as the layers go deeper while the shortcut connection provide a fix towards to potential ``feature collapse'' issue. To this end, additional branches of augmented shortcut is proposed to further improve the feature diversity as well as the final recognition power.

**Limitations And Societal Impact:**

Limitations:
- The motivation of this work is straightforward, from the observation that the rank, i.e. diversity, is lost when layers go deeper. However, this phenomenon does not appear merely in vision transformers. Instead, this issue has also be explored in modern CNNs structures. Based on this observation, lots of progress has been made, such as Channel Pruning [1], compression [2] and semantic segmentation [3]. Meanwhile, the lost-rank phenomenon of self-attention layer is originally investigated in [4], which this work is based on. Thus the contribution of this paper might not be adequate.

- The technical novelty of the proposed module is also limited. There have been some prior works that transform/enhance the shortcut pathway. It's quite obvious that adding a re-parameterized shortcut branch is able to improve the performance. Furthermore, another concern is about the memory cost brought by additional shortcuts.

[1] Channel Pruning for Accelerating Very Deep Neural Networks. ICCV 2017

[2] On compressing deep models by low rank and sparse decomposition. CVPR 2017

[3] Tensor Low-Rank Reconstruction for Semantic Segmentation. ECCV 2020

[4] Attention is not all you need: Pure attention loses rank doubly exponentially with depth. arxiv preprint

**Main Review:**

- The authors propose to augment the shortcuts in vision transformers. This technique is independent from the specific architecture designs, being able to be plugged into different vision transformers.

- The demonstration of feature diversity is clear and interesting. The motivation of augmenting the shortcut is well explained through the figures and tables in Section 2.

- This paper is well-written. All detailed techniques are clearly clarified. Extensive experiments are conducted to validate the significance of the proposed simple module.

- Weakness and Limitation: see next section.

---------------Post Rebuttal Feedback-------------
After reading the authors' rebuttal, some of my concerns are solved. Thus I increase my rating to 6.

**Time Spent Reviewing:**

8 hours

---

> ### Author Response · Authors · 2021-08-10
> **Our response to Reviewer 1wm5**
>
> Thanks for your constructive comments.
>
> **Q1**: The motivation of this work is straightforward.
>
> **A1**: Thanks a lot for your concise summary of our observation that the rank, i.e., diversity is lost when layers go deeper in vision transformers. But in our humble opinion, this observation does not widely hold in modern CNNs structures. For example, we measure the diversity of channels in a plain CNN model (VGG-16) and show the results in Table R1. It shows that the diversity metric $r(Z_l)$ does not decline along with the increase of network depth. Note that VGG-16 does not have shortcuts but can still achieve 71.6% top-1 accuracy on ImageNet [r5]. While for vision transformers with more than 4 blocks, removing the shortcuts will incur extremely low accuracies (Table 1 in the main paper). Their features from different patches will become almost the same in deep layers (Figure 3 (a) and Figure 4), and we believe this phenomenon of vision transformers is an important factor that causes the performance difference.
>
> Also, the suggested works ([r1], [r2]) about channel pruning and compression were motivated by the low-rank assumption over filters, rather than the decreased rank along with the increasing of network depth. Actually, [r1] shows that more redundancy exists in shallower layers instead of deeper layers (Figure 4 in r1). [r3] is an interesting work that designs a low-rank-to-high-rank context reconstruction framework to model the 3D representations, which achieves high performance on semantic segmentation tasks. We will include more discussions on this aspect in the final version.
>
> Specifically, [r4] is a pioneering work to understand the self-attention layer and analyzes the role of shortcut in standard architectures, e.g., BERT, Albert and XLNet. But the lost-rank issue has not been fully explored in vision transformers yet, let alone the development of a solid solution to address the lost-rank. In this paper, we develop the parameterized shortcuts to enhance the feature diversity in vision transformers. Theoretical and empirical analyses are given to demonstrate the effectiveness of the proposed augmented shortcut.
>
> Table R1: Diversity metric $r(Z_l)$ varies w.r.t. layers in CNNs (VGG-16 on ImageNet).
>
> | Layer                               | 1    | 5    | 10   | 13   |
> | ----------------------------------- | ---- | ---- | ---- | ---- |
> | Diversity metric $r(Z_l)$ | 0.67 | 0.73 | 0.71 | 0.78 |
>
> **Q2**: The technical novelty.
>
> **A2**: We think identifying the performance bottleneck of modern vision transformers and the intuition behind the simple augmented shortcuts should be put in the first place. The proposed parameterized shortcuts to address the feature diversity result from our in-depth reflection on the vision transformers. Figure 3 and Figure 4 show that vision transformers suffer from feature collapse and the proposed augmented shortcuts can significantly improve the feature diversity, especially in deep layers. Theorem 2 further analyzes the effectiveness of augmented shortcuts from a theoretical perspective. By improving the feature diversity, this simple, efficient module achieves obvious performance improvement (about 1% top-1 accuracy) over various vision transformer architectures (e.g, DeiT, PVT, T2T). Compared with the re-parameterized methods (e.g., [r6]) that fuse multiple branches into one path at the deployment phase, the proposed augmented shortcut skips a whole MSA or MLP block together, which cannot be simply merged.
>
> **Q3**: The memory cost of the proposed method.
>
> **A3**: Thanks for the nice concern. For efficiency, we develop a portable scheme that implements the parameterized shortcuts with block-circulant projections. The memory required for storing extra weight parameters can be ignored (Line 189-198 in the main paper). Moreover, we further test the GPU memory occupancy of different models on a V100 GPU with batch-size 1024. The proposed Aug-ViT-S model occupies 8413MiB memory, which is only 0.5% larger than that of the ViT(DeiT)-S model (8373MiB).
>
> [r1] Channel Pruning for Accelerating Very Deep Neural Networks, ICCV 2017.
>
> [r2] On compressing deep models by low rank and sparse decomposition, CVPR 2017.
>
> [r3] Tensor Low-Rank Reconstruction for Semantic Segmentation, ECCV 2020.
>
> [r4] Attention is not all you need: Pure attention loses rank doubly exponentially with depth, arxiv preprint.
>
> [r5] https://pytorch.org/vision/stable/models.html.
>
> [r6] ACNet: Strengthening the Kernel Skeletons for Powerful CNN via Asymmetric Convolution Blocks, ICCV 2019.

---

### Official Review · Reviewer_KRcF · 2021-07-19

**Rating:** 6
**Confidence:** 3

**Summary:**

This paper observes the problem of feature collapse in transformer, where the tokens in deep layers collapse to the same representation. It then proposes to add additional augmented shortcuts in parallel to the original shortcut connection in the multihead self-attention modules and MLP modules, based on the observation that simple identity shortcut alleviates the problem of feature collapse. In the experiments, the proposed method brings 1.1% performance gain in the ImageNet classification task compared to the baseline model. Performance gains can also be seen from COCO detection task and other downstream finegrained classification tasks, as compared to the baseline models.

**Limitations And Societal Impact:**

See above. The societal impact is shown one the last page of the manuscript.

**Main Review:**

In this paper, a simple add-on approach is proposed to encourage the feature diversity in deep transformer layers. It introduces additional shortcut connections, comprising of a linear layer and an activation non-linearity, on top of the existing shortcuts. The feature diversity is measured by the matrix norm of the difference with the best rank-1 approximation matrix to the feature matrix. This diversity measurement is improved compared to both the setting without any shortcut or with identity shortcut(Figure 3 and Figure 4).

Circulant martices are used as the weights of the linear layers introduced in the additional shortcuts to save computation and storage. As compared to using the full randomly initialized weight matrices, this approach uses roughly 1/200 the storage and 1/20 the G-FLOPs, with minimum performance drop.

The main modification centers around the shortcut connections in transformer architecture. Therefore it would be more convincing if more ablations could be done around the theme of shortcuts. Specifically, how does the choices of stochastic depth/dropout parameters influence the performance? How does the architecture of the shortcuts(e.g. multiple linear layers, choices of non-linearity, etc.) affects the diversity of the token feature outputs?

Moreover, the paper observes the decrease in feature diversity as the depth of the model increases, and that adding more shortcut connections can effectively solve this issue. An experiment with deeper transformer model, such as the CaiT family, is therefore much sought for. It would be more convincing if the paper can show improved diversity in token features of deeper transformer model.

------------------------------------------------Post Rebuttal--------------------------------------------------------------------------

The responses have addressed the issues listed in the previous comment, so I will  I will increase the final score to 6.

**Time Spent Reviewing:**

3.5 hours

---

> ### Author Response · Authors · 2021-08-10
> **Our response to Reviewer KRcF**
>
> Thanks for your constructive comments.
>
> We conduct more ablations following your suggestions and show the results below. More ablation studies and detailed analyses will be included in the final version.
>
> **Q1**: How does the choice of stochastic depth/dropout parameters influence the performance?
>
> **A1**: We change the parameters of stochastic depth/dropout for both ViT(DeiT)-S model and Aug-ViT-S, and the results are shown in Table R1 and R2. We empirically find the default parameters for DeiT (drop rates of stochastic depth/Dropout are set as 0.1/0.0) also work well for the Aug-VIT models. In addition, the models equipped with augmented shortcuts tend to have higher tolerance to over-large drop rates. For example, with drop rate=0.3 for dropout, the accuracy of Aug-ViT-S can still achieve 75.1% accuracy, while the accuracy of DeiT-S decreases to 69.8%.
>
> Table R1: Top-1 accuracy (%) varies w.r.t. the parameters of stochastic depth.
>
> | Drop rate of stochastic depth | 0    | 0.05 | 0.1  | 0.2  | 0.3  |
> | ----------------------------- | -----| -----| ---- | -----| -----|
> | ViT(DeiT)-S                   | 79.7 | 79.8 | 79.8 | 79.3 | 78.6 |
> | Aug-ViT-S                     | 80.9 | 80.9 | 80.9 | 80.4 | 80.0 |
>
> Table R2: Top-1 accuracy (%) varies w.r.t. the parameters of dropout.
>
> | Drop rate of Dropout | 0    | 0.05 | 0.1  | 0.2  | 0.3  |
> | -------------------- | ---- | -----| -----| -----| -----|
> | ViT(DeiT)-S          | 79.8 | 78.9 | 77.3 | 74.0 | 69.8 |
> | Aug-ViT-S            | 80.9 | 80.3 | 79.1 | 76.5 | 75.1 |
>
> **Q2**: How does the architecture of the shortcuts (e.g., multiple linear layers, choices of non-linearity, etc.) affects the diversity of the token feature outputs?
>
> **A2**: Thanks for your suggestion. We further conduct experiments to investigate the impact of shortcut architecture.
>
> 1) Table R3 shows how the accuracies of Aug-ViT-S vary w.r.t. the number of linear layers on ImageNet. For the Aug-ViT-S model with two augmented shortcuts, 'X-X' denotes the number of linear layers in each path. We observe that better performance is achieved when the two paths have different depths (e.g, 1-3), implying that more diverse features could be produced.
>
> 2) For the non-linearity, we replace the default GeLU with other wide-used activation functions (e.g., PReLU, ReLU) and the results are shown in Table R4.  Compared with the baseline DeiT-S with 79.8% accuracy, performance improvements are achieved with these three activation functions. Especially, GeLU is an effective activation function used in transformers, which also achieves the best performance here.
>
> Table R3: The number of layers in augmented shortcuts.
>
> | The number of linear layers | 1-1         | 2-2         | 1-2         | 1-3        |
> | --------------------------- | ----------- | ----------- | ----------- | ---------- |
> | Aug-ViT-S                   | 80.9 (+1.1) | 80.8 (+1.0) | 81.0 (+1.2) | 81.1(+1.3) |
>
> Table R4: The choices of non-linear activation functions.
>
> | Non-linear activation function | GeLU        | PReLU       | ReLU        |
> | ------------------------------ | ----------- | ----------- | ----------- |
> | Aug-ViT(DeiT)-S                | 80.9 (+1.1) | 80.7 (+0.9) | 80.5 (+0.7) |
>
> **Q3**: Experiments with deeper transformer models.
>
> **A3**: We have conducted experiments on a 24-layer T2T-24 model, which is much deeper than the DeiT family with 12 layers (Table 2 in the main paper).
>
> Following your nice suggestions, we implement CaiT model [r1] to validate the effectiveness of augmented shortcuts on deeper models. Specifically, we select a 36-layer CaiT-XXS-36 model as the baseline and then train an Aug-CaiT-XXS-36 model using the proposed method. The results of these two models are in Table R5, which shows that the proposed augmented shortcut approach still works well on deeper transformer models, which brings about 1% increment regarding to the top-1 accuracy. The variety of diversity metric $r(Z_l)$ w.r.t. layers is shown in Table R6, which shows the feature diversity is obviously improved. Experiments on more models will be included in the final version.
>
> Table R5: Performance of CaiT-XXS-36 and Aug-CaiT-XXS-36 on ImageNet. The resolution of input image is 224x224.
>
> | Model           | Top-1 Accuracy (%) |
> | --------------- | ------------------ |
> | CaiT-XXS-36     | 79.1               |
> | Aug-CaiT-XXS-36 | 80.1               |
>
> Table R6: The diversity metric $r(Z_l)$ w.r.t. layers in CaiT-XXS-36 and Aug-CaiT-XXS-36.
>
> | Layer                        | 1    | 10   | 20   | 30   | 36   |
> | -----------------------------| ---- | ---- | ---- | ---- | ---- |
> | CaiT-XXS-36 without shortcut | 0.61 | 0    | 0    | 0    | 0    |
> | CaiT-XXS-36                  | 0.90 | 0.85 | 0.81 | 0.75 | 0.71 |
> | Aug-CaiT-XXS-36              | 0.91 | 0.88 | 0.86 | 0.85 | 0.83 |
>
> [r1] Going deeper with Image Transformers, arxiv preprint.

---

### Decision · Program_Chairs · 2021-09-27

**Decision:**

Accept (Poster)

**Comment:**

This paper got mixed ratings initially. All the reviewers agree with the motivation and effectiveness of the proposed method, but they also raise the concerns on the novelty and technique contributions of this work. In particular, the feature collapse issue has been identified by a prior work [8]. The theorem listed in the paper was not developed by this work either. The reviewers also suggested to add more ablation experiments on this work. After the response, the authors address these concerns and all the reviewers give positive ratings. AC has read the submission, reviews and authors' response. Although the finding of feature collapse is not firstly identified by this work, the proposed shortcut augmentation is simple and the performance improvement is significant. Thus, AC agrees with reviewers that this work would be valuable for the community and recommend acceptance.